# Engineering selectivity of *Cutibacterium acnes* phages by epigenetic imprinting

**Nastassia Knödlseder[1], Guillermo Nevot[1], Maria-José Fábrega[1], Julia Mir-Pedrol[1], Marta Sanvicente-García[1], Nil Campamà-Sanz[1], Bernhard Paetzold[2], Rolf Lood[3], Marc Güell[1]***

**1** Department of Medicine and Life Sciences, Universitat Pompeu Fabra, Barcelona, Spain, **2** Sbiomedic, Beerse, Belgium, **3** Division of Infection Medicine, Department of Clinical Sciences, Lund University, Lund, Sweden

\* marc.guell@upf.edu

**Data Availability Statement:** The datasets supporting the conclusions of this article are available in the European Nucleotide Archive under project number PRJEB42527. Custom analysis

## Abstract

*Cutibacterium acnes (C. acnes)* is a gram-positive bacterium and a member of the human skin microbiome. Despite being the most abundant skin commensal, certain members have been associated with common inflammatory disorders such as *acne vulgaris*. The availability of the complete genome sequences from various *C. acnes* clades have enabled the identification of putative methyltransferases, some of them potentially belonging to restriction-modification (R-M) systems which protect the host of invading DNA. However, little is known on whether these systems are functional in the different *C. acnes* strains. To investigate the activity of these putative R-M and their relevance in host protective mechanisms, we analyzed the methylome of six representative *C. acnes* strains by Oxford Nanopore Technologies (ONT) sequencing. We detected the presence of a 6-methyladenine modification at a defined DNA consensus sequence in strain KPA171202 and recombinant expression of this R-M system confirmed its methylation activity. Additionally, a R-M knockout mutant verified the loss of methylation properties of the strain. We studied the potential of one *C. acnes* bacteriophage (PAD20) in killing various *C. acnes* strains and linked an increase in its specificity to phage DNA methylation acquired upon infection of a methylation competent strain. We demonstrate a therapeutic application of this mechanism where phages propagated in R-M deficient strains selectively kill R-M deficient acne-prone clades while probiotic ones remain resistant to phage infection.

## Author summary

Bacteria have several mechanisms to prevent infection by bacteriophages. One of them introduces modifications to its own DNA to differentiate own from non-own. Some strains of the skin bacterium *C. acnes* were analyzed for their ability to introduce these modifications to their own genome, and one was found to be able to do so. We could show that bacteria without the ability to modify their own DNA were infected strongly by the bacteriophage while the modified bacteria were immune to the infection. The skin is highly diverse in its composition of certain *C. acnes* strains, and it is believed that a change

scripts for the methylation and SLST phage analysis are freely available in https://bitbucket.org/synbiolab/guell_methylation/src/master/.

**Funding:** This work was funded by the Office of Naval Research (Award N62909-18-1-2155), INNOValora (INNOV21-09-1 - SynFlora) given by Universitat Pompeu Fabra and Industria del Coneixement of the Catalan Government (AGAUR - IdC 2019 PROD 00057), all granted to MG. NK is funded by Maria Maetzu-UPF fellowship (AEI- MM-CEX2018-000792-M D.COMAS - PROGRAMA ESTATAL DE FOMENTO INVESTIGACIÓN CIENTÍFICA Y TÉCNICA DE EXCELÉNCIA - UNIDADES DE EXCELÉNCIA MARIA DE MAEZTU 2018) and by the Sociedad Española de Químicos Cosméticos (Beca de la SEQC para la presentación de trabajos en el 33rd IFSCC Congress - KNÖDLSEDER, NASTASSIA). GN is supported by the Secretariat for Universities and Research of the Ministry of Business and Knowledge of the Government of Catalonia and the European Social Fund with an FI 2021 grant (Award 2021FI_B1 00128). MJF is funded by a Juan de la Cierva Fellowship from the Spanish Government (Award FJC 2018-037096-I). NCS is funded by ONRx for the project Engineered Biofilms, with Modular Functionality for Persistent and Survivable naval Platforms (Award N62909-20-1-2086). The funders had no role in study design, data collection and analysis, decision to publish, or preparation of the manuscript.

**Competing interests:** I have read the journal's policy and the authors of this manuscript have the following competing interests: MG and BP are founders and shareholders of S-Biomedic

in abundance of certain strains can cause disease. The use of antibiotics for example usually kill bacteria regardless of their strain type and therefore does not protect the symbiotic relationship of the human with its microbiome. So, we tested if we could selectively kill a certain sub-population of strains by making the phage target only strains without the ability to modify their own DNA. We show that we could modulate the composition of strains over time and we could save strains with beneficial traits while reducing acne-prone strains.

## Introduction

In the natural environment, prokaryotes exchange DNA through horizontal gene transfer (HGT). This transfer is regulated through mechanisms like Restriction Modification systems (R-M) or CRISPR (clustered regularly interspaced palindromic repeats) to protect the hosts lineage specificity from exogenous DNA or phage infection [1]. In fact, ~90% of sequenced bacterial genomes harbor a R-M system [2–4] usually comprising two components: a Methyltransferase (MTase) and a Restriction Endonuclease (REase). The MTase methylates endogenous DNA by transferring a methyl group to either the N6-methyladenine (6mA), the N4-methylcytosine (4mC) or the N5-methylcytosine (5mC) to discriminate against foreign incoming DNA. Then, the REase cleaves specific, unmodified patterns recognized as foreign at their phosphodiester bonds [5]. There are four different types of R-M system depending on sequence recognition, cleavage position, cofactor necessity and substrate specificity, which have been described in depth by J.R. Roberts *et al*. 2003 [5,6]. Understanding R-M functionality is key to achieve efficient genetic engineering of bacteria. Indeed, R-M is one of the main causes of genetic intractability of non-model bacteria in which the system enables bacteria to distinguish their own DNA from foreign, invading genetic elements [2].

Although DNA methylation in bacteria was discovered half a century ago [7], the recent advent of next generation sequencing technologies has enabled the high-throughput study of bacterial methylomes. In particular, Oxford Nanopore Technologies (ONT) sequencing identifies base modifications by comparing their current signal to the signal yield by a canonical base as they go through a protein pore. In a similar fashion, Single Nucleotide Real Time (SMRT) sequencing uses the kinetics by which new bases are incorporated to determine if a base is modified or not.

The gram-positive *C. acnes* is the most abundant skin commensal of the human skin microbiome but has been associated with the skin condition acne vulgaris [8]. However, since *C. acnes* is the major component of healthy skin its concrete involvement in the disease is not fully understood [9]. Despite being found in the human skin, it is also a major contaminant in post-surgery infections [10–12]. *C. acnes* shows only little genetic variation between different clades and comparative genomic analysis of a genome subset showed that only 7–11% of CDS differs from the core genome. However, Island-like genomic regions and mobile genetic elements were found to have been horizontally acquired [13–15]. Additionally, linear plasmids have been found in both type I and type II *C. acnes* strains which might be acquired through conjugative transfer, encoding virulence factors like the tight adherence (tad) locus [12,16].

*C. acnes* genome stability might be related to inhabiting a highly specific ecological niche where it outcompetes many other bacterial species [17,18] or/and due to defense strategies which would not allow genetic changes [19]. Indeed, different defense strategies have been found within certain strains. For example, *C. acnes* clade II strains contain full CRISPR/Cas type I-E systems with spacers matching phages while clade I has lost big parts over time,

indicating that they represent a more recent evolutionary lineage compared to clade II strains [13]. Putative R-M systems have been described in REBASE but only one active system has been identified up to date: a type IIIB system recognizing the motif 5'-AGCAGY-3'[20]. In addition, the described linear plasmid contains a toxin-antitoxin system which was previously proposed to be synonymous in their function to R-M systems by regulating and limiting the genetic flux between lineages [21].

C. acnes harbors multiple temperate and lytic phages. Phages within the skin modulate skin microbiome composition by infecting either a narrow range of strains or broadly within a population [22]. Interestingly, a healthy microbiome harbors more phages than an unhealthy one [23]. While C. acnes phages have an overall sequence identity of more than 85%, their ability to infect certain strains is very diverse [24] and the reason for this diversity is not well understood. One of the most studied C. acnes bacteriophages is PAD20 which belongs to the group of Siphoviruses and was determined to neither be a strictly lytic nor a strictly lysogenic phage; determined by the absence of stable lysogens and absence of genes related to integration [25,26]. It was suggested that PAD20 might follow a pseudolysogenic life cycle and might persist as a non-replicative extrachromosomal plasmid [25]. PAD20 phage shows a general broad host range but has been shown to be less lytic against C. acnes belonging to clade II strains [22].

In the current study, we use Nanopore ONT and SMRT sequencing to study the methylome of six different C. acnes strains in order to further characterize their putative R-M systems. All six newly sequenced strains belong to three clade types IA, IB and clade II which were clustered based on their distinct evolutionary lineage by McDowell et al. into phylotypes [27,28]. While most of the previous typing schemes are based on multiple gene sequences (multiple-locus sequence typing: MLST) [27–33], Scholz et al. presented a single-locus sequence typing (SLST) scheme with a comparable resolution to MLST which reflect the phylogenetic strains of C. acnes with a single locus and which we have used here for sub-classification of strains [34].

Additionally, we assessed the importance of R-M systems in the defense of C. acnes against foreign plasmid DNA and its natural phages in order to shed light on how these systems remain functional or not. We demonstrate how we can efficiently modulate host selectivity. For instance, we produce phages lacking epigenetic imprints to target acne-prone strains such as A1 (clade IA) while protecting probiotic strains such as H1 (clade IB) [35].

## Results

### Restriction-methylation in *C. acnes* strains

To understand host protective mechanisms in C. acnes we investigated the presence and activity of R-M systems in a selection of strains. We used curated HMM models of restriction-modification protein motifs found in PFAM and the dedicated tools available in REBASE to find potential R-M systems in all our C. acnes laboratory strains based on homology to available genome assemblies (for a full list of strains and accession numbers we refer to S3 Table).

Comparing genomes of S3 Table, three main potential R-M systems were found in C. acnes clades I and II which include R-M IIB, IIG and IIIB (Fig 1A).

The R-M IIB system contains an orphan methylase with an ORF of 1251 bp. Interestingly, a S- adenosyl synthase gene is located just upstream of the methylase gene. Regarding type IIG R-M systems in C. acnes, differences have been observed depending on the clade. Type IIG has been described to be inactivated in C. acnes IB strains by the insertion of a 30 kb pro-phage between PPA1578 and PPA1614 and by a deletion of 4 genes in C. acnes clade II strains. C. acnes clade IA, on the other hand, have an intact reading frame or have mutations leading to a premature stop codon which generates two independent reading frames (Fig 1A) [14]. For

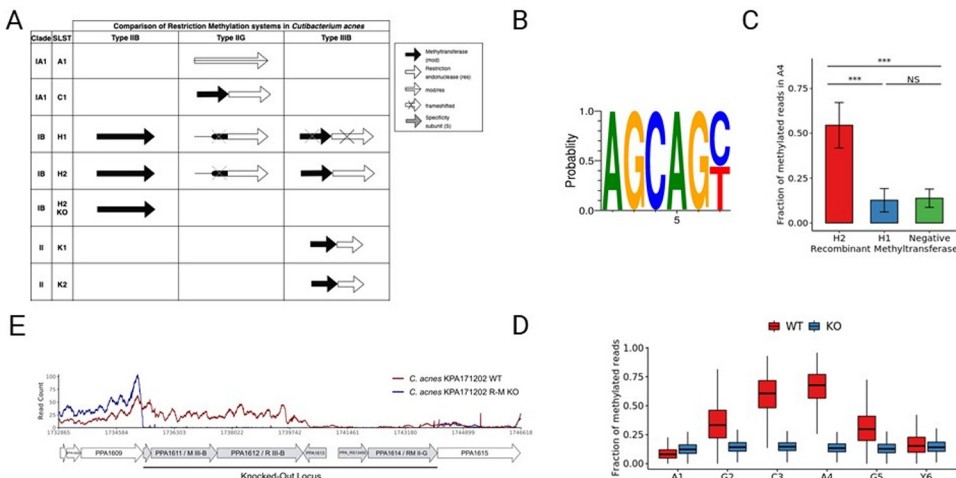

**Fig 1. *C. acnes* KPA171202 (SLST H2) has a functional R-M system with a methylation motif in the AGCAGY sequence.** - (A) Genomic comparison of potential R-M systems in selected *C. acnes* strains using assemblies available in REBASE and comparing strain or clade level. Black represents the MTase while white represents the REase. (B) Sequence logo of the methylation motif from the reads with highest methylation score. (C) Fraction of methylated reads in the methylated adenine of the AGCAGY motif detected in the pMW535 plasmid after recombinant expression of H1 or H2 IIIB methyltransferase. (D) Fraction of methylated reads across the positions of the AGCAGY motif for the wild-type (red) and the type IIIB R-M system knock-out (blue) KPA171202 (H2) strain. (E) Transcriptional profile of restriction methylation locus and surroundings in *C. acnes* KPA171202 Wild Type (SLST H2 WT) and *C. acnes* restriction methylation knock-out (SLST H2 R-M KO). Expression levels are displayed by RNA-seq read counts per genomic position (mean of two replicates). R-M IIG corresponds to PPA1614 and R-M IIIB to PPA1611.

example, one single frame is detected in SLST A1 while two frames are detected in SLST C1 (Fig 1A). R-M IIIB contains a methylase (Mod) and a restriction endonuclease (Res) as two independent ORFs. Our laboratory strain SLST H1 has a premature stop codon resulting from a two base pair deletion in the Mod gene and deletions in the Res gene resulting in an altered frame which probably compromises its functionality (Fig 1A). Interestingly, Clade IB (such as SLST H1 and H2) strains have three predicted restriction modification systems while clade IA (such as SLST A1 and C1) have only one (Fig 1A). This could be explained due to the evolution of clade IA which gained an increased benefit over other strains [14].

Sequence alignment of clade II *C. acnes* strains with the R-M type IIIB system in clade IB strains, show sequence similarities but also suggest that in type II *C. acnes* the type IIIB R-M system has been partially deleted (Fig 1A).

## *De novo* motif discovery of representative *C. acnes* strains

To get an overview of the functionality of the different putative ORFs, we chose representatives from different clades (Fig 1A) and conducted ONT sequencing for *de novo* motif discovery. For clade IA we chose SLST A1 and C1, for clade IB we chose SLST H1 and H2 (KPA171202) [36], for clade II we chose SLST K1 (09–9) [12] and K2.

Out of all these strains, only H2 showed a significant methylation motif corresponding to the AGCAGY consensus sequence, in which the methylated base corresponds to the adenine in fourth position (Fig 2A). However, not all sites found in SLST H2 were detected as methylated. As the Tombo package does not suggest a fixed cut-off for classifying whether a base is modified or not, we inferred a cut-off based on the density of the modified fraction of reads for the methylated position in the motif. In order to do so, we calculated the mean of the right peak (0.67) and defined the cutoff as the mean minus two standard deviations, establishing it

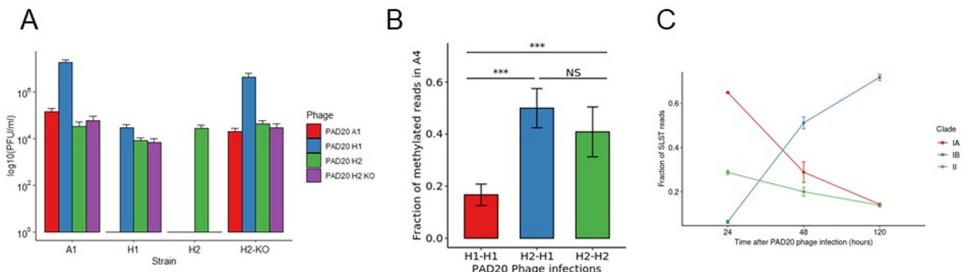

**Fig 2. R-M system IIIB of *C. acnes* KPA171202 affects PAD20 Phage infection properties and protects the bacteria from lysis.** - (A) Variation of infection properties of PAD20 phage propagated on either SLST A1, H1, H2 or H2 KO. (B) Fraction of methylated reads in the methylated adenine at 4th position of the AGCAGY motif in the PAD20 phage genome. Samples are named first after the infected strain where the phage was sequenced and second after the strain on which the phage was previously propagated (i.e, H2-H1 means a phage produced in H1 was used to infect H2 and then extracted for sequencing from H2) Significance was tested using ANOVA analysis (***, p < 0.001). (C) Selected killing of whole microbiome samples infected with PAD20 propagated on SLST A1 after 3 different time points. Samples were taken after 24, 48, 120 h of on-going PAD20 infection. Samples were amplified using SLST methodology and subjected to Illumina sequencing. Similarity to SLST alles is used to classify sequencing reads into clades IA, IB and II. Clade IA contains following SLST types: A1, A2, A3, A4, A7, A11, A13, A16, A17, A18, A21, A24, A27, A28, A29, A30, A31, A33, A34, A35, A38, A40, A42, A46, A47, A48, A49, A51, A52, C1, C5, F4, F6, F10, F15, F16, F17, F22, F24; Clade IB contains: H1, H2, H3, H4, H7, H8, H10, H11, H12; Clade II contains: K1, K2, K7, K9, K17, K19.

at 0.39. Using this criterion, the frequency of methylated motifs was 94%. Additionally, fine typing using a multi-label classification algorithm called Nanodisco showed the modification to be present at the N6 of the second adenine in the motif (AGC**A(6mA)**GY) (S1C Fig).

To further confirm the lack of methylation patterns, we performed SMRT sequencing in two of the tested strains corresponding to SLST A1 and H1. In accordance with our ONT sequencing data, none of these two strains showed a methylation motif (S1A and S1B Fig).

## Linking IIIB Mod gene to MTase functionality

To understand the methylation properties of the IIIB methylases from H1 and H2 we recombinantly expressed those in *E. coli* dam- dcm- strain together with a plasmid pMW535 containing eight AGCAGY recognition sites [37]. In the case of SLST H2 MTase, the ONT sequencing on the plasmid revealed the same AGCAGY methylation pattern as found in the genome (Fig 1C). This confirms that the pattern observed at the genome level is due to the activity of this protein. However, the recombinant expression of the disrupted ORF proceeding from the H1 MTase did not show any clear methylation pattern nor the characteristic methylation at AGCAGY despite having potential homologies to the same type IIIB system as the H2 strain (Fig 1C and 1A).

On the other hand, we wanted to test the restriction properties of SLST A1, H1 and H2 strains in degrading the unmethylated plasmid in a cell extract. As expected, degradation of the plasmid occurred only with the H2 strain lysate. Neither A1 nor H1 lysates were able to digest the unmethylated plasmid, matching our previous analysis which predicts no active R-M system for these two strains (S1D Fig).

To further link the Mod gene to the methylation activity we created a *C. acnes* KPA171202 methylation deficient strain by deleting a 9 kb locus including both R-M IIIB and R-M IIG system and inserting an erythromycin resistance cassette for selection similarly to Sörensen *et al.* 2010 [37] (Fig 1A). We verified correct knockout by junction PCR and WGS followed by a functional characterization using ONT sequencing and RNA sequencing of WT vs. KO strain (Figs 1D and 1E and S2D). ONT sequencing verified the loss of methylated AGCAGY motifs

in absence of MTase in comparison to the wildtype genome (Fig 1D). In addition, RNA sequencing showed no expression of the IIIB and IIG locus in the KO strain (Figs 1E and S2D). Also, we did not detect expression of the IIB and IIG R-M systems in the WT strain corresponding to our previous findings using ONT sequencing, where no specific patterns were detected and linked to those R-M systems (Fig 1E).

## Identification of host range specificity of PAD20 phage propagated on various strains

After identifying a functional R-M system in SLST H2 (KPA171202), we hypothesized a link between phage infection properties and the availability of active R-M systems in *C. acnes*. In this study, we propagated PAD20 *C. acnes* phage on SLST strains A1, H1, H2 and H2 KO. We used those phages from different origins to infect a representative selection of *C. acnes* strains of every clade and measured its ability to form lysis plaques. Those belonging to clade IA, in particular SLST A1, A5, F4, C1, C3, were lysed by all four versions of the PAD20 phages (Figs 2A and S1E). Notably, clade IB strains followed different patterns of infection.

PAD20 A1 showed the least infection properties by only being able to lyse clade IA and SLST H2 KO but leaving SLST H1 and H2 unaffected. Interestingly, SLST H2 KO became sensitive to PAD20 A1 infection which might be linked to the loss of protection due to the knocked-out R-M IIG system since SLST H1 still having R-M IIG was not sensitive to PAD20 A1. PAD20 H1 had a broader lysis profile by lysing clade IA, SLST H2 KO, SLST K1 and itself. PAD20 H2 showed extended lysis potential by lysing all clade IA and clade IB strains and one clade II strain, SLST K1 (Figs 2A and S1E). PAD20 H2 KO showed the same phenotype in infection as PAD20 H1, which verifies the loss of protective properties linked to the R-M IIIB system. On the other side, it became sensitive to PAD20 A1 which could be linked to the lost R-M IIG system. Furthermore, SLST K2 strain (clade II) was not able to be lysed by any of the three phages even though the phage originally was extracted from clade II (Figs 2A and S1E). This is probably due to the existence of CRISPR spacers against PAD20 phages found in its genomic DNA. From 13 spacers found in K2, 6 aligned against PAD20 (S1 Table). Additionally, we showed expression of the complete CRISPR/Cas type I-E locus in K1 and K2 by RNA-seq analysis (S2A and S2B Fig) and verified expression levels of genes belonging to the CRISPR system by comparing threshold cycle ($C_t$) obtained by each candidate gene in both strains tested (S2C Fig).

## Restriction-methylation modulates infection potential of PAD20 phages

In order to validate our observation and hypothesis that epigenetic imprinting of phage genomic DNA leads to differences in host infectivity we tested if the PAD20 phage inherited the methylation pattern after infection. PAD20 genomic DNA contains 20 sites with the recognition motif AGCAGY. We extracted the phage DNA after infection of different strains and sequenced them using ONT sequencing. As expected, only phages propagated in H2 showed the characteristic AGCAGY motif methylated conversely to phage produced in other strains (Fig 2B). Additionally, we observed that phages initially propagated on H1, followed by a propagation on H2 had detectable AGCAGY methylation patterns (Fig 2B).

## Selective killing of acne-prone strains in a whole microbiome sample

Since we observed that the phage origin plays a role in the ability of lysis, we wondered if we could take advantage of this effect to selectively target certain strains in a complex sample. We cultured together two *C. acnes* isolates SLST A1 and H1 and infected them with phage PAD20 propagated on SLST A1. We randomly selected 10 single colonies and sequenced those using

the SLST scheme. We verified that all tested colonies showed the same SLST type H1, proving A1 was selectively lysed in the co-culture.

To further test our hypothesis, we extended the previous experiment to a whole microbiome culture extracted from human skin. The complex bacterial population was cultured together with PAD20 A1 phage and the progressive change in proportion of different strains through illumina sequencing at 3 different time points (24, 48, 120 h) was analyzed. We compared changes on clade level, since phages propagated on A1 had similar lysis profiles on all tested members of the same clade (S1E Fig). Comparing the changes in proportion of certain clades over time, we observed a clear decrease in Clade IA, while Clade IB gained in proportion (Fig 2C). In case of type II strains, differences on strain level within the clade were expected since spacers against PAD20 phages were identified previously in diverse type II isolates (S1 Table) [13,26]. A slight decrease of *C. acnes* clade II strains could be observed but overall stayed quite stable over time. This resistance to phage infection is potentially due to the presence of active type I-E CRISPR systems with matching spacers against PAD20 phage in some of the strains in the clade II population (Fig 2C).

## Discussion

In this report, we describe one functional R-M system in *C. acnes* and assess its impact on host range sensitivity of phage PAD20. We tested the functionality of different putative R-M systems present in different clades of *C. acnes*. In particular, we studied three putative systems corresponding to R-M IIB, IIG and IIIB, all shown to be present either in the available genome assemblies of *C. acnes* or in our laboratory strains. We have shown the IIIB system in SLST H2 (KPA171202) to be the only functional one and detected its specific AGCAGY recognition motif in genomic, plasmid and phage DNA through ONT sequencing, being consistent with previous findings [20].

In addition, we observed this system's involvement in foreign DNA defense as a cell extract from SLST H2 was able to digest an unmethylated plasmid. These findings could explain, together with the limited experimental data available, the genetic intractability of this microbe.

For SLST A1 and H1 no functional R-M system was detected, nor through ONT or SMRT sequencing. Several efforts have been taken to classify *C. acnes* strains using its sensitivity to phage infection [22,38]. However, most strains have been shown to be sensitive, showing little success in these approaches. As mentioned before, this study shows that infectivity depends on which host the phage has been propagated on. This fact could explain the differences in these typing studies where the phages were isolated only from their host or from a specific propagating strain.

In addition, we have shown that R-M systems play a role in the defense of infection in *C. acnes* as the H2 KO (ΔR-M IIIB, IIG) increased the strain sensitivity to PAD20 phage by losing its protective mechanism and therefore acquiring a lysis profile similar to A1 but still being able to lyse H1. Furthermore, we should consider that other undescribed defense mechanisms in *C. acnes* might also act as infectivity regulators. In whole, the differential infectivity should not only be regarded in laboratory cultures but also in the dynamical skin environment. In the skin, strains are constantly lysed producing new phages with different host-range specificities. This potentially represents an ecological regulatory mechanism, especially considering the abundance of *C. acnes* phages in the skin[39]. Nonetheless, further studies need to clarify whether this effect plays a major role in natural environments.

Finally, this finding uncovers the possibility to shape and engineer the infectivity range of *C. acnes* phages for phage therapy. We showed that epigenetic imprinting of phage DNA can change the susceptibility of phages on their host and can change specificity in infection. This

opens the door for developing phage cocktails that could potentially modulate strain composition on human skin.

## Conclusions

In conclusion, we hypothesize that R-M plays a big part in specific strains of *C. acnes* defensive strategies. While in H2 strain KPA171202 evidence for a defensive mechanism was identified, in other *C. acnes* strains it remains unclear. For Type II strains like K1 and K2 it has been claimed that they might be less evolved due to the findings of complete CRISPR/Cas type I-E systems while in other subtypes like IA and IB only remaining parts were found [13]. In clade IA we believe other defense mechanisms might be available since there are no publications to date that prove successful genetic engineering in those clades.

Nevertheless, identifying R-M in *C. acnes* will not only enhance engineering efforts by either circumventing restriction sites, in-vivo methylating DNA, eliminating those sequences from targeting vectors or knocking out available functional systems but also allows us to program selective killing of specific *C. acnes* strains by epigenetic imprinting.

## Methods

### Homology-based search of potential restriction methylation systems

In order to identify the potential R-M systems present in *C. acnes*, we first retrieved from the REBASE database all the predicted ORFs corresponding to either methyltransferases or restriction enzymes from those strains of which complete genome assembly was available. For the strains with no information available, we retrieved their respective genomes from the NCBI Genome database and performed protein-protein BLAST with the annotated coding sequences to identify the potential R-M homologs. To further assess the difference between systems across strains, we performed pairwise comparisons between ORFs belonging to the same putative system by sequence alignment using the Clustal Omega algorithm available at EMBL-EBI portal [40]. Refer to S3 Table for a complete list of strains and accession numbers used in this study.

Additionally, we used dedicated HMM models for well-known phage-defense systems using DefenseFinder[41,42] as well as manual HMMER (v3.3) *hmmsearch* queries with downloaded models from PFAM database[43]. In the case of K2, we used CRISPR-CAS Finder (https://crisprcas.i2bc.paris-saclay.fr/CrisprCasFinder/Index) to further characterize the CRISPR genes and locus identified. The obtained spacers were aligned against the reference genome of PAD20 with blastn.

### Genomic DNA extraction

*C. acnes* SLST (single locus sequence typing) A1, C1, H1, H2, K1, K2 were grown on Brucella agar plates for 3 days. Bacteria were collected with a sterile cotton swab and inoculated in 1 mL BHI media, spun down 3 min 8000xg and resuspended in 1 mL lysis buffer (9,4 parts TE buffer pH 8,0 and 0,6 parts 10% SDS). All content was transferred to tubes prefilled with 0.1 mm silica beads and lysed in a mechanical homogenizer (Savant Bio 101 FastPrep FP120 Cell disruption system) for 25 sec at 6,5 speed. Samples were placed on ice for 1 min and the cycle was repeated once more. Next, 6 ul Proteinase K (20 mg/mL) and 5 ul RNAse (1 mg/mL) were added and incubated 1 h at 37°C water bath. Equal volume Phenol/Chloroform was added, mixed and spun down at max speed for 5 min. Then, the aqueous phase was collected and the previous step repeated. Equal volume of chloroform was added, mixed and spun down at max speed for 5 min. The aqueous phase was collected and 2,5 volumes of 100% ice-cold EtOH was added to precipitate DNA. Samples were kept for at least 1 h at -20°C. Then, they were spun

down at max speed for 15 min at 4˚C. Supernatant was discarded and the pellet was washed with 70˚C EtOH. After, the sample was spun down for 2 min at max speed. Once dried, the pellet was resuspended in 50 ul H20 and DNA concentration was measured using Qubit HS DNA kit (Thermo Fisher).

### Bacterial lysate preparation

*C. acnes* strains were pre-cultured in 20 ml BHI broth with 180 rpm shaking at 37˚C during 24 hours under anaerobic conditions. Then, they were re-inoculated in 1 L of fresh BHI with 1200 rpm of stirring, maintaining the same conditions. After 72h, cells were harvested by centrifugation at 5000 xg for 20 minutes and the bacterial pellet was washed twice with 20 ml of cold $K_2PO_4$ 50mM, pH 7.4. Then, 0.33 ml of S30 buffer (10 mM Tris($CH_3COO$) (pH 8.2), 14 mM Mg($CH_3COO$)$_2$, 10 mM K($CH_3COO$) and 4 mM DTT) were added per gram of wet cell mass and the suspension was transferred into 2ml microtubes prefilled with 0.1 mm glass beads. Using a mechanical homogenizer (Savant Bio 101 FastPrep FP120 Cell disruption system), cells were lysed following 2 cycles at 6000 rpm for 30 seconds. Samples were then filtered by centrifugation at 1000 xg for 5 minutes at 4˚C using a 0.22 um filter unit. Then, the follow-through was centrifuged at 12.000 xg for 12 minutes at 4˚C. Finally, clear supernatants were collected and stored at -80˚C until they were used.

### *C. acnes* lysate assay with plasmid DNA

*C. acnes* lysates were incubated with 0.5 ug plasmid DNA together with different buffers (NEB) supplemented with or without ATP or SAM. Cutsmart (NEB) buffer was chosen for further experiments. 8 ul of *C. acnes* concentrated lysate was mixed with 2 ul of Cutsmart buffer (1x final) and 0,5 ug of plasmid DNA was added. Final volume was adjusted to 20 ul and samples were incubated for 4 h at 37˚C. Afterwards, DNA was observed on a 1% agarose gel.

### *C. acnes* PAD20 Phage infection

*C. acnes* SLST types A1, H1, H2 (KPA171202) and restriction-methylation deficient H2 (H2 KO) were infected with a high titer of *C. acnes* PAD20 phage extracted from either A1 or H1. Brucella top agar plates (0,6%) were prepared by mixing 4,5 mL Brucella agar with 0,5 mL Bacterial culture. After being solidified, phages were added in 10 ul drops and plates were incubated at 37˚C anaerobically for 3–5 days. After the first infection cycle, phages were extracted by adding 10 mL SM buffer (100 mM NaCl, 8 mM $MgSO_4$, 50 mM Tris-HCl pH 7.5, 0.01% gelatin) to the plates and incubating 30 min at RT carefully rotating. SM buffer was collected and NaCl was added to a final concentration of 0.5 M to release phages bound to bacterial debris. Samples were incubated for 30 min at RT and centrifuged at 8000g for 20 min. Then, samples were filtered and 10% (w/v) PEG 6000 was added. After 10 minutes of incubation, the phages were collected by centrifugation (10,000 xg, 30 min). Finally, the pellet was resuspended in 300ul of SM buffer. Phage concentration was determined by serial dilutions followed by plaque counting. EOP assay was performed by using an initial phage concentration of $5\times10^3$ PFU/mL to re-infect *C. acnes* A1, A5, C1, C3, F4, H1, H2 (KPA171202), K1 (09–9) and K2 in serial dilutions. Plaquing efficiency of phages PAD20 A1, PAD20 H1, PAD20 H2 and PAD20 H2 KO was measured in PFU/mL and compared to each other.

### In vitro phage selectivity testing

C. acnes SLST types A1 and H1 were mixed 1:1 and inoculated to an OD600 of 0,1. A volume of 50 ul of PAD20 phage ($3\times10^6$ PFU/mL) was added and phage infection proceeded for 3

days anaerobic growth, 37˚C, 110 rpm. After 3 days serial dilutions were plated on Brucella agar plates, grown for 3 days anaerobically at 37˚C and a subset of 10 colonies characterized by SLST PCR and subsequent Sanger sequencing.

## Whole microbiome phage infection assay and strain genotyping

Whole microbiome was extracted by swapping as previously described in Paetzold et. al 2019 [44]. Bacteria was grown on agar plates anaerobically for 5 days prior to inoculation into BHI media to enrich in *C. acnes*. Inoculation was done to a $OD_{600}$ of 0,1 and 50 ul of PAD20 extracted from A1 (3x10^6 PFU/mL) was added. The culture was grown until time point 120h while samples were taken after 24h, 48h and 120h. To extract DNA 10 ul of culture was added to 500 ul QuickExtract Solution (Epicentre) and treated 65˚C for 6 min followed by 98˚C for 2 min. 5 ul of this solution was used for amplification of the SLST region linked to illumina adaptors using KAPA HiFi HotStart Readymix (Roche) (Initial denaturation for 5 min at 95˚C followed by 35 cycles of 98˚C for 20 s, 62˚C for 25 s, and 72˚C 30 s; and a final elongation for 1 min at 72˚C).

SLST_illumina_fwd: 5'-**TCGTCGGCAGCGTCAGATGTGTATAAGAGACAG**TTGCTC GCAACTGCAAGCA-3' and

SLST_illumina_rev: 5'-**GTCTCGTGGGCTCGGAGATGTGTATAAGAGACAG**CCGGC TGGCAAATGAGGCAT -3'.

Samples were PCR purified and afterwards barcoded with Nextera XT indices using KAPA HiFi HotStart Readymix (Initial denaturation 95˚C for 3 minutes followed by 8 cycles of 95˚C for 30 seconds, 55˚C for 30 seconds, 72˚C for 30 seconds and a final elongation of 72˚C for 5 minutes) for sequencing in a single Illumina flow cell. Illumina MiSeq was conducted and an adequate depth reserved per sample. Hamming distance between MiSeq paired-end reads of the same length and all SLST alleles were calculated with DistanceMatrix from DECIPHER (https://rdrr.io/bioc/DECIPHER/man/DistanceMatrix.html). Each read was assigned to the most similar strain, taking into consideration all alleles with the same minimal distance. Since there were several sequences with the same distance to more than one strain, we grouped similar strains. After obtaining all distances, reads were filtered allowing a maximum of 3 mismatches between the paired reads and the SLST alleles, because this was the minimal distance to discriminate between the established groups without uncertainty. Finally, we have calculated the proportion between different clades (IA, IB, and II) at each time point (24, 48, and 120 hours) using the number of paired reads with a minimal distance with a certain strain or with several strains of the same clade.

## Creating a methylation deficient KPA171202

We followed a strategy previously described by Sörensen *et al.* 2010 [37] to create a methylation deficient KPA171202 strain. Shortly, we amplified 500 bp upstream (primers 468/469) and downstream (primers 470/471) of the locus to be replaced and cloned those into the pGEM-T-easy vector. In a second step the erythromycin cassette was cloned between the homology arms using Acc65I restriction sites and positive clones selected to be transformed into dam- GM2199 cells. *C. acnes* competent cells were prepared as previously described and transformants selected on Brucella plates containing 10 ug/mL erythromycin [37]. Knock-out was verified by WGS and junction PCR using primer pairs 499/338 and 308/500.

## PCR amplification, cloning and recombinant type IIIB methylase expression

Methylase genes corresponding to type IIIB were amplified using KAPA HiFi HotStart Polymerase (Roche) and primer pair 283 and 284 (KPA-MIIIb_fwd and rev) to include the

constitutive promoter BBa_J23100 (S2 Table). PCR products were TOPO cloned using the Zero Blunt TOPO cloning kit (Life TECHNOLOGIES) and transformed into dam-dcm- *E. coli* strain (New England BioLabs) and co-transformed with pMW535, a *C. acnes* suicide vector containing eight AGCAGY motifs. A single colony was grown overday in 5 mL LB broth and then inoculated in 200 mL o/n. Afterwards, plasmid DNA was extracted using HiPure Maxiprep kit (Life TECHNOLOGIES).

## Preparation of Nanopore libraries

Genomic libraries corresponding to *C. acnes* A1, H1, H2, C1, K1, K2 strains, PAD20 phage DNA and *E. coli* plasmids were prepared using the Ligation Sequencing Kit SQK-LSK109 (Oxford Nanopore) according to the manufacturer's instructions. Native and WGA samples were barcoded using the Native Barcoding Kit (EXP-NBD104) and PCR Barcoding kit (EXP-PBC001), respectively. All samples were sequenced on the MinION platform either using a FLO-MIN106D (R9) flow cell or a FLO-FLG001 Flongle (R9.4.1) flow cell. Basecalling and demultiplexing were performed using Guppy v4.0.11.

## Data analysis and de novo motif detection

Nanopore sequencing data for the mentioned strain were analyzed using Tombo v1.5 software [45] for detection of modified bases. In brief, current signals were compared against the KPA171202 reference genome using the "resquiggle" command. Then, methylation peaks were detected comparing the native sample and the whole genome amplification (WGA) sample for each strain using the "detect_modifications model_sample_compare" command. For the detection of methylated motifs, first we extracted the 50 bp context of 1000 positions with the highest modified fraction (*i.e.* the fraction of reads identified as potentially methylated over all reads mapping to the given position in the genome) using the "text_output signif_sequence_context" command. This set of sequences was processed using MEME [46] with 'zoops' mode to identify common sequences. Statistically significant modified fraction values for each genomic position were exported from tombo using tombo's "text_output browser_files" and were further analysed using a custom R script (see code availability). A similar procedure was used to analyse the Phage DNA and plasmid samples. For phage DNA, the genome reference was assembled *de novo* using Fyle [47] with the default parameters. Fine typing of the methylation motif was performed using the multi-label classification algorithm present in the Nanodisco [48] toolkit according to its documentation.

## SMRT library preparation and sequencing

Library was prepared using Pacific Biosciences protocol for SMRTbell Libraries using PacBio Barcoded Adapters for MultiplexSMRT Sequencing. The library was sequenced on Pacific Biosciences Sequel instrument using Sequel Polymerase v2.1, SMRT cells v2 LR and Sequencing chemistry v2.1. This sequencing service was provided by the Norwegian Sequencing Centre (www.sequencing.uio.no).

## RNA extraction, cDNA synthesis and qRT-PCR

Total RNA was extracted using RNeasy Mini kit with RNAprotect Bacteria reagent (Qiagen) and treated with RNase-free DNase I (Qiagen) on column. RNA was quantified using a Nano-Drop One spectrophotometer (Thermofisher). 500 ng of RNA were used to perform the RT-PCR in random hexamer primed reactions using RevertAid First strand cDNA synthesis kit (Thermofisher). qRT-PCR was carried out in a QuantStudio 12K Flex Real-Time PCR

System (Applied Biosystems). Each reaction contained 25 ng transcribed cDNA, 0.5 uM each fwd and rev primer (S2 Table) and 1x PowerUp SYBR Green Master Mix (Applied Biosystems) following manufacturer's recommendation. Three independent replicates were made for each sample and gene.

## RNAseq

Isolated RNA purity and integrity was assessed using Bioanalyzer (Agilent Technologies GmbH, Germany). Library preparation was done with Truseq Stranded Total RNA and samples were sequenced using Illumina (Macrogen, South Korea) at 60 million paired reads.

RNA-seq analysis was performed using the nf-core RNA-seq pipeline v3.0 [49,50] in Nexflow v20.12.0-edge [51]. Raw paired-end reads were trimmed using Trim Galore v0.6.6 and aligned to *C. acnes* strains H2, K1 or K2 genomes using STAR v2.6.1d and SAMtools v1.10 [52–54]. Quality control was performed using FastQC v0.11.9 [55]. Mapped reads were counted using *mpileup* from BCFtools in htslib v1.11 [56].

## Supporting information

**S1 Fig.** (A) (B) Distribution of the Modification QV value from SMRT sequencing across different coverage regions for SLST H1 (A) and SLST A1 (B) samples. No clear enrichment is shown for any of the bases, indicating the lack of a methylation motif. (C) Predicted modification for the different positions within the AGCAGY motif according to the multi-label classification algorithm. Most modifications were classified as 6mA and located in the 4th adenine of the motif. (D) Restriction profiles of plasmid DNA incubated with lysates of *C. acnes* SLST H2 (KPA171202), H1 and A1. Lane 1 and 9 molecular weight markers 1 kb GeneRuler (Thermo Fisher); Lane 2, plasmid control pMW535 1ug; Lane 3, H2 lysate; Lane 4, H2 lysate+1ug pMW535; Lane 5, H1 lysate; Lane 6, H1 lysate+1ug pMW535; Lane 7, A1 lysate; Lane 8, A1 lysate+1ug pMW535 (E) PAD20 phage infection properties on different *C. acnes* strains depending on propagation origin (SLST A1, H1, H1 or H2 KO). (E) Infection of a selection of *C. acnes* strain belonging to different clades with PAD20 propagated on SLST A1, H1, H2 and H2 KO. Phages acquire different infection properties depending on their origin.
(TIF)

**S2 Fig.** Transcriptional profile and architecture of CRISPR/cas type I-E locus found in (A) *C. acnes* SLST K1 (09–9) and (B) SLST K2. Expression levels are displayed by RNA-seq read counts per genomic position. Black diamonds and grey squares represent CRISPR clusters, with the number of total repeat-spacer-repeats indicated below. (C) Expression levels of genes associated with the CRISPR cascade in *C. acnes* SLST K1 (blue) and K2 (red) determined by qRT-PCR across 3 independent replicates per strain and gene. RecA was used as control for a housekeeping gene. Displayed are threshold cycles ($C_T$) and were used to classify gene expression as high (21–25 $C_T$), Moderate (26–30 $C_T$) or Low (31–38 $C_T$). Error bars indicate standard error. (D) Top panel: Expression levels in *C. acnes* KPA171202 Wild Type (SLST H2 WT) and restriction methylation locus knockout (SLST H2 R-M KO) displayed by logarithm of RNA-seq transcripts per million plus one (TPM+1) (mean of two replicates) per gene. Bottom panel: Restriction methylation operon architecture showing knocked out loci.
(TIF)

**S1 Table. SLST K2 strain CRISPR spacers against PAD20 phage.**
(TIF)

**S2 Table. Table of primers used in the study.**
(XLSX)

**S3 Table. Complete list of *C. acnes* strains and GeneBank accession numbers used in this study.**
(XLSX)

## Acknowledgments

We want to express our deepest gratitude to Dr. Holger Brüggemann for providing initial plasmids (pMW535) and strain 09–9 (K1) and many helpful discussions and advice. We would also like to thank Dr. Javier Santos Moreno and Dr. Amal Rahmeh for critically reviewing the manuscript.

## Author Contributions

**Conceptualization:** Nastassia Knödlseder, Marc Güell.

**Data curation:** Nastassia Knödlseder, Guillermo Nevot, Maria-José Fábrega.

**Formal analysis:** Nastassia Knödlseder, Guillermo Nevot, Julia Mir-Pedrol, Marta Sanvicente-García, Nil Campamà-Sanz.

**Funding acquisition:** Marc Güell.

**Investigation:** Nastassia Knödlseder, Guillermo Nevot, Maria-José Fábrega.

**Methodology:** Nastassia Knödlseder, Bernhard Paetzold, Rolf Lood, Marc Güell.

**Project administration:** Nastassia Knödlseder, Marc Güell.

**Resources:** Marc Güell.

**Software:** Guillermo Nevot, Julia Mir-Pedrol, Marta Sanvicente-García, Nil Campamà-Sanz.

**Supervision:** Marc Güell.

**Validation:** Nastassia Knödlseder, Guillermo Nevot, Maria-José Fábrega.

**Visualization:** Nastassia Knödlseder, Guillermo Nevot, Julia Mir-Pedrol, Marta Sanvicente-García, Nil Campamà-Sanz.

**Writing – original draft:** Nastassia Knödlseder, Guillermo Nevot.

**Writing – review & editing:** Nastassia Knödlseder, Guillermo Nevot, Maria-José Fábrega, Julia Mir-Pedrol, Marta Sanvicente-García, Nil Campamà-Sanz, Bernhard Paetzold, Rolf Lood, Marc Güell.

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
