## [Editor Report · Decision Letter 0]

1 Feb 2022

Dear Dr. Guell,

Thank you very much for submitting your manuscript "Engineering selectivity of Cutibacterium acnes phages by epigenetic imprinting" for consideration at PLOS Pathogens. As with all papers reviewed by the journal, your manuscript was reviewed by members of the editorial board and by independent reviewers through Review Commons. In light of the reviews, we would like to invite the resubmission of a significantly-revised version that takes into account the reviewers' comments.

We cannot make any decision about publication until we have seen the revised manuscript and your response to the reviewers' comments. Your revised manuscript is also likely to be sent to reviewers for further evaluation.

Sincerely,

Michael Wessels

Section Editor

PLOS Pathogens

Kasturi Haldar

Editor-in-Chief

PLOS Pathogens

orcid.org/0000-0001-5065-158X

Michael Malim

Editor-in-Chief

PLOS Pathogens

orcid.org/0000-0002-7699-2064
---

## [Decision Letter · Decision Letter 1]

7 Mar 2022

Dear Dr. Güell,

We are pleased to inform you that your manuscript 'Engineering selectivity of Cutibacterium acnes phages by epigenetic imprinting' has been provisionally accepted for publication in PLOS Pathogens.

Best regards,

Michael Wessels

Section Editor

PLOS Pathogens

Kasturi Haldar

Editor-in-Chief

PLOS Pathogens

orcid.org/0000-0001-5065-158X

Michael Malim

Editor-in-Chief

PLOS Pathogens

orcid.org/0000-0002-7699-2064

Reviewer Comments (if any, and for reference):

Reviewer's Responses to Questions

**Part I - Summary**

Reviewer #1: The Authors have addressed all my concerns. Congratulations for their work!

**Part II – Major Issues: Key Experiments Required for Acceptance**

Reviewer #1: (No Response)

**Part III – Minor Issues: Editorial and Data Presentation Modifications**

Reviewer #1: (No Response)

PLOS authors have the option to publish the peer review history of their article (what does this mean?). If published, this will include your full peer review and any attached files.

Reviewer #1: **Yes: **Pedro H. Oliveira

---

## [Editor Report · Acceptance letter]

24 Mar 2022

Dear Dr. Güell,

We are delighted to inform you that your manuscript, "Engineering selectivity of Cutibacterium acnes phages by epigenetic imprinting," has been formally accepted for publication in PLOS Pathogens.

Best regards,

Kasturi Haldar

Editor-in-Chief

PLOS Pathogens

orcid.org/0000-0001-5065-158X

Michael Malim

Editor-in-Chief

PLOS Pathogens

orcid.org/0000-0002-7699-2064